

# Annual memory in the terrestrial water cycle

Wouter R. Berghuijs[1], Ross A. Woods[2], Bailey J. Anderson[3,4], Anna Luisa Hemshorn de Sánchez[1], Markus Hrachowitz[5]

[1]Department of Earth Sciences, Free University Amsterdam, Amsterdam, The Netherlands
[2]School of Civil, Aerospace, and Design Engineering, University of Bristol, United Kingdom
[3]WSL Institute for Snow and Avalanche Research SLF, Swiss Federal Institute for Forest, Snow and Landscape Research WSL, Davos Dorf, Switzerland
[4] Institute for Atmospheric and Climate Science, ETH Zurich, Zurich, Switzerland
[5]Department of Water Management, Delft University of Technology, Delft, Netherlands

*Correspondence to*: Wouter R. Berghuijs (w.r.berghuijs@vu.nl)

**Abstract.** The water balance of catchments will, in many cases, strongly depend on its state in the recent past (e.g., previous days). Processes causing significant hydrological memory may persist at longer timescales (e.g., annual). The presence of such memory could prolong drought and flood risks and affect water resources over long periods, but the global universality,
strength, and origin of long memory in the water cycle remain largely unclear. Here, we quantify annual memory in the terrestrial water cycle globally using autocorrelation applied to annual time series of water balance components. These timeseries of streamflow, global gridded precipitation, GLEAM potential and actual evaporation, and a GRACE-informed global terrestrial water storage reconstruction indicate that, at annual timescales, memory is typically absent in precipitation but strong in terrestrial water stores (rootzone moisture and groundwater). Outgoing fluxes (streamflow and evaporation)
positively scale with storage, so they also tend to hold substantial annual memory. As storage mediates flow extremes, such memory also often occurs in annual extreme flows and is especially strong for low flows and in large catchments. Our model experiments show that storage-discharge relationships that are hysteretic and strongly nonlinear are consistent with these observed memory behaviours, whereas non-hysteretic and linear drainage fails to reconstruct these signals. Thus, a multi-year slow dance of terrestrial water stores and their outgoing fluxes is common, it is not simply mirroring precipitation memory,
and appears to be caused by hysteretic storage and drainage mechanisms that are incorporable in hydrological models.

## 1 Introduction

Temporal variability of weather, (subsurface) hydrological processes, and land-surface conditions can cause streamflow to vary by orders of magnitude over time, leading to droughts and floods, and affecting water resources for society and ecosystems (Van Loon, 2015; Blöschl et al., 2020; Berghuijs et al., 2017; Kreibich et al., 2022). Temporal autocorrelation in streamflow
timeseries tends to be strong at short (e.g. daily) timescales because streamflow often strongly depends on catchment storage (e.g., Kirchner, 2009; Spence et al., 2010; McNamara et al., 2011; Riegger and Tourian, 2014) which, compared to precipitation, changes relatively gradually with time (e.g., Lischeid et al., 2021; Li et al., 2024). At longer timescales (e.g., one



year or longer), streamflow can also be autocorrelated (e.g., Mudelsee, 2007; Godsey et al., 2014). This autocorrelation may
be interpreted as evidence of the persistent influence of the prior hydroclimatic state of a catchment on current hydrologic
conditions. Signals of past climate, which are detectable in streamflow records for long periods, are known as "catchment-
state persistence", or "catchment memory" (de Lavenne et al., 2022; Orth and Seneviratne, 2013). Catchment memory has the
potential to prolong the risks of droughts (Sutanto and Van Lanen, 2022), may lead to recorded flood-rich and flood-poor
periods (Blöschl et al., 2020), and could affect water availability for society and ecosystems (Klemeš et al., 1981). These long
and short-term components of water storage often largely determine how streamflow responds to precipitation and, for
instance, may cause different hydrologic responses to the same precipitation amounts (e.g., Kirchner, 2024).

Long memory, defined here as a signal which persists for more than one year, is not uncommon in streamflow records, but it
is unclear whether, and to what extent, this characteristic is present across catchments. Following its discovery in annual flow
records of the River Nile (Hurst, 1951), more widespread evidence of long memory in annual flows has been reported for
many rivers worldwide (e.g., Mudelsee 2007; Labat, 2008; Kanthelhardt et al., 2006; Montanari, 2003; Markonis et al., 2018;
Szolgayova et al., 2014; Milly and Dunne, 2002; de Lavenne et al., 2022).  Low flows and floods have been reported to
sometimes be significantly clustered in time (Gudmundsson et al., 2011; Blöschl et al., 2020; Lun et al., 2020; Sutanto and
Van Lanen, 2022), suggesting the possibility of memory effects even for streamflow extremes. However, globally, it remains
mostly unquantified how universal long memory is across the water cycle and to what extent it occurs in extreme flows.

Mass conservation dictates that streamflow memory cannot occur in full isolation but must interact with other water balance
components:

$$\frac{dS}{dt} = P - Q - E \qquad\qquad\qquad (1)$$

In this example catchment water balance (ignoring potential inter-basin groundwater flows), $S$ is storage, $P$ is precipitation, $Q$
is streamflow, $E$ is evaporation, and $t$ is time. This continuity equation implies that long memory in one component is physically
connected with other components. Such connections have also been demonstrated previously. For example, Milly and Dunne
(2002) show that, for several catchments, interannual carry-over of storage cascades into the memory of annual flows. Godsey
et al. (2014) shows how, in several catchments, seasonal low flows are significantly correlated to both the current and the
previous year's snowpacks. Yet, in most cases, such connections remain largely unexplored, as most large-scale assessments
of streamflow memory do not simultaneously analyse other water balance components (e.g., Mudelsee 2007; Labat, 2008;
Kanthelhardt et al., 2006; Montanari, 2003; Markonis et al., 2018; Szolgayova et al., 2014).

Long memory in hydroclimatological variables has been quantified beyond streamflow. For example, precipitation can be
affected by large-scale multi-year climate oscillations (e.g., El Niño, North Atlantic Oscillation, Pacific Decadal Oscillation)
(Ropelewski and Halpert, 1987; Jong et al., 2016). Such persistence of states in the ocean-atmosphere system (e.g. ENSO,





Pacific Decadal Oscillation) can be a contributor to multi-annual persistence in precipitation and streamflow (Devon et al., 2004; McKerchar, and Henderson, 2003; Ward et al., 2014) but global analysis suggests that significant memory in annual precipitation is uncommon (Sun et al., 2018). Instead in regional and semi-global studies, autocorrelation rapidly declines in precipitation timeseries (Markonis and Koutsoyiannis, 2016; Kantelhardt et al., 2006; Fraedrich and Blender, 2003; Potter, 1979). While evapotranspiration is the second-largest terrestrial water flux (Dorigo et al., 2021), its long memory has not been

systematically quantified globally.

Terrestrial liquid water storage mostly consists of groundwater and soil moisture. Aquifers in arid regions can exhibit long-term memory as they integrate precipitation variabilities, the effects of which persist over long periods (e.g., Opie et al., 2020). Groundwater often strongly fluctuates seasonally (Strassberg et al., 2007), but long timeseries can also exhibit multi-year

drying or wetting trends (e.g., Rodell et al., 2018; Jasechko et al., 2024) that likely hold significant long memory. Furthermore, Gravity Recovery and Climate Experiment (GRACE) data can be used to forecast trends in global land water storage for the following year (Li et al., 2024) indicating significant long memory. Nevertheless, long groundwater memory remains mostly undocumented at global scales. Soil moisture storage memory has been mostly analysed for shorter timescales (e.g., Orth & Seneviratne, 2012; Martínez-Fernández et al., 2021), while its long memory remains unclear globally.


Here we simultaneously quantify the annual memory of different water balance components to reveal where this memory originates, and how it cascades between processes. We assess to what extent this memory occurs in global precipitation, temperature, soil moisture, total terrestrial water storage, streamflow, and evaporation timeseries. We subsequently use empirical and model tests to establish which catchment factors shape the observed memory behaviours.

## 2 Methods

### 2.1 Global data

We use *streamflow* data from 15029 catchments from the Global Streamflow Indices and Metadata Archive (GSIM), a worldwide collection of metadata and indices derived from streamflow time series (Do et al., 2018; Gudmundsson et al., 2018). Gridded *precipitation* timeseries are from the Global Precipitation Climatology Centre (GPCC) V7 0.5° data set for 1981–

2020 (Becker et al., 2013; Schneider et al., 2008), and 2m global land surface temperature from 1981–2020 at 0.5° from the Global Historical Climate Network (GHCN)/Climate Anomaly Monitoring System (CAMS) (Fan & van den Dool, 2008). We also use 0.25° GLEAM V3.8a estimates of *actual evaporation*, *potential evaporation*, and *rootzone soil moisture* over the period 1981–2020 (Martens et al., 2017). In GLEAM, potential evaporation is based on the Priestley and Taylor equation, and actual evaporation is based on potential evaporation multiplied by an evaporative stress factor based on observations of

microwave Vegetation Optical Depth (VOD) and estimates of root-zone soil moisture (Martens et al., 2017). We use long-term (i.e., 1981–2022) 0.25° resolution monthly *terrestrial water storage* estimates over the land surface from GTWS-MLrec





(Yin et al., 2023), which is a reconstruction that uses a set of machine-learning models with several predictors and GRACE (Yin et al., 2023). We exclude data from Greenland, Iceland, and Antarctica from our analysis.

## 2.2 Annual memory and autocorrelation

We quantify memory as autocorrelation of annual values (mean, minimum, maximum) of hydrological fluxes or stores, at a one-year lag time (Fig. 1). To compute autocorrelation, we linearly detrend timeseries of annual values and then calculate autocorrelation $\rho_y$ at lags of one year as:

$$\rho_y = \frac{\text{Cov}(y_t, y_{t-1})}{\sqrt{\text{Var}(y_t) \cdot \text{Var}(y_{t-1})}} \qquad (2)$$

$\rho_y$ measures the linear correlation between the detrended time series of variable $y$ at year $t$ and of that variable at year $t$-1.  $\rho_y$

can range from -1 to 1, where larger positive values indicate a stronger tendency for adjacent values to be similar. We calculate $\rho_y$ for time series with at least 20 years of continuous data. We test the probability of accepting the null hypothesis that the residuals $y_t$ are uncorrelated ($p$-value<0.1 for statistical significance, consistent with Sun et al., 2017).

A strong autocorrelation may not be hydrologically significant if it is associated with small volumes of water relative to other

terms in the water balance. Therefore, we also express memory in terms of their water volumes. These *memory volumes $V_y$* can be calculated based on the autocorrelation $\rho_y$. Namely, $\rho_y$ is essentially the equivalent to the slope of a linear regression between $y_t$ and $y_{t-1}$ and thereby also expresses an elasticity of $y_t$ to $y_{t-1}$:

$$\rho_y = \frac{\text{Cov}(y_t, y_{t-1})}{\sqrt{\text{Var}(y_t) \cdot \text{Var}(y_{t-1})}} \approx \frac{\mathrm{d}y_t}{\mathrm{d}y_{t-1}} \cdot \frac{\overline{y_{t-1}}}{\overline{y_t}} \approx \frac{\mathrm{d}y_t}{\mathrm{d}y_{t-1}} \qquad (3)$$

Thus, $\rho_y$ expresses the unit change in $y_t$ associated with a unit change in $y_{t-1}$. To calculate the mean memory volume, we

calculate the mean absolute annual anomaly of a variable $y$, and multiply this with the autocorrelation of $\rho_y$.

$$V_y = \rho_y \cdot \overline{|z_y|} \qquad (4)$$

where $V_y$ is the mean annual memory volume of process $y$, and $\overline{|z_y|}$ is the mean absolute annual anomaly of process $y$. Calculating $V_y$ for several water balance components (e.g., *S, P, E, Q*) allows tracing, in terms of water volumes, how much long memory on average occurs in different components of the water cycle.




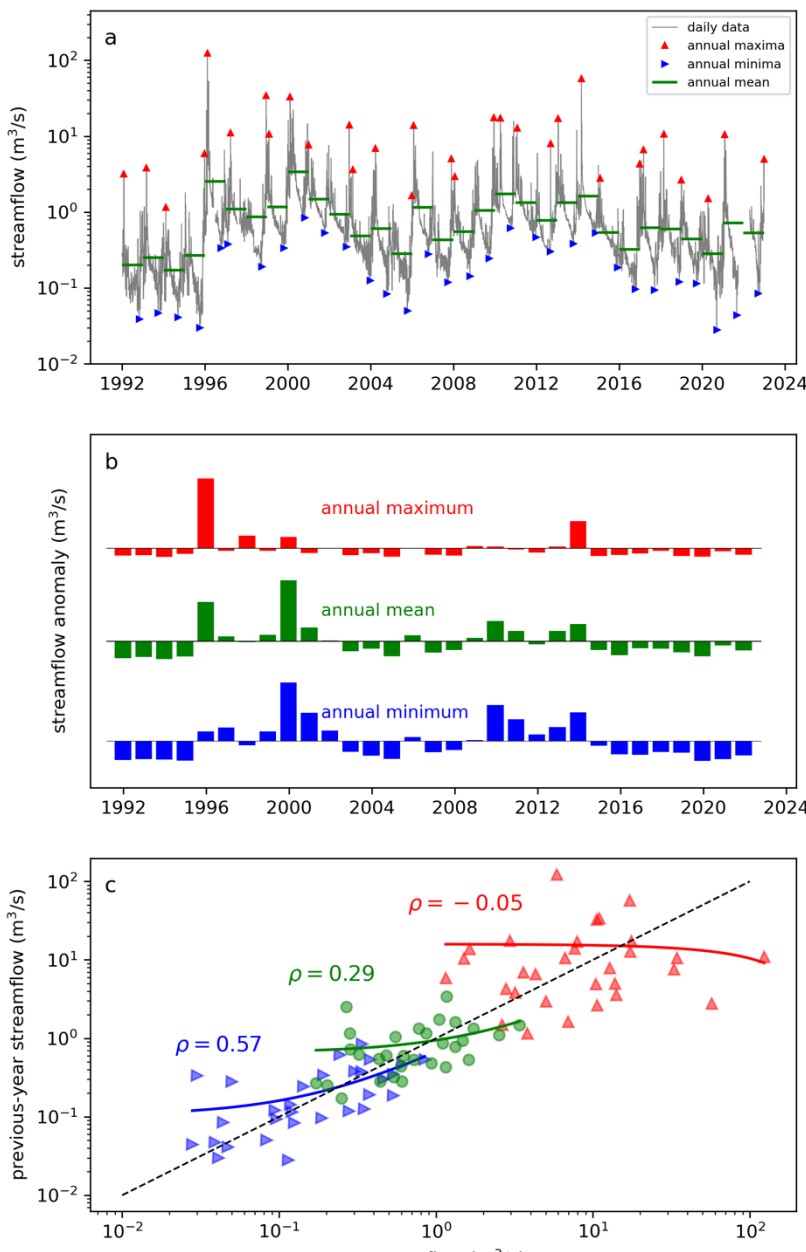

**Figure 1.** Example streamflow timeseries of the Suidkaap River in South Africa (25.73 °S, 30.98 °E) highlighting the memory in annual mean, minimum, and maximum flows. From these daily timeseries, annual streamflow indices (mean, maximum, minimum) are selected (a). The associated annual anomalies of these streamflow indices (i.e., annual deviations from their mean value) already suggest that years with above- (or below-) average rates tend to follow one another (b). Correlation coefficients between each year's streamflow and the previous year's streamflow indices quantify the autocorrelation of the mean (0.29), minimum (0.57), and maximum (-0.05) annual flow rates. Note that these correlations are based on linear regressions, but these linear regressions appear non-linear on logarithmic axes. For this river, annual mean and annual minimum flows hold substantial memory, whereas annual maximum flows are largely uncorrelated (and even show some, statistically insignificant, negative autocorrelation).



## 3 Results and discussion

### 3.1 Memory of water balance components

Globally, terrestrial precipitation tends to have no significant annual memory (Fig. 2a), as its autocorrelation is on average weak (mean $\rho_P$ = 0.006), and only 12% of grid cells globally have significant positive autocorrelation at a 90% confidence interval (i.e., *p*-value <0.10). This pattern of overall weak autocorrelation is in line with earlier analysis that reports autocorrelation as statistically significant for ~14% of the global land surface (Sun et al., 2018). The absence of systematic long memory in precipitation timeseries does not indicate that long memory never occurs, as, for example, the El Niño/Southern Oscillation (ENSO), has widespread effects on regional precipitation (Ropelewski & Halpert, 1987). However, the effects of such multi-year climate cycles appear to not result in widespread annual memory in precipitation time series globally. Due to this weak precipitation memory (Fig. 2a), its memory volume also tends to be small (Fig. 2b). Memory volumes are below 5 mm/year in over 97% of the grid cells, indicating that only a small water volume statistically relates to the previous year's precipitation rate. Thus, while precipitation is the water source of other water balance components, its memory (as indicated by both the weak autocorrelation and the small associated memory volumes) is unlikely the direct cause of systematic annual memory elsewhere in the water cycle.

Annual memory in water balance components can also originate from memory of other atmospheric conditions that drive evaporation, such as potential evaporation or temperature, but neither shows particularly strong long memory. GLEAM (Priestley-Taylor based) potential evaporation tends to have some more memory (mean $\rho_{E_P}$ = 0.09) than precipitation and this memory is significant for 26% of the land surface (Fig. 2c). The annual variability of potential evaporation tends to be relatively small (compared to precipitation variability) thus the potential water volumes associated with it are also generally small in volume (Fig. 2d). Air temperature tends to have weak autocorrelation (mean $\rho_T$= 0.01; not shown here). The overall weaker memory in climatic forcing compared to evaporation and streamflow (shown below) suggests that annual memory elsewhere in the water cycle does not simply mirror memory in forcing timeseries and therefore also originates from other processes.

More substantial annual memory arises globally once precipitation accumulates in storage. Both rootzone soil moisture (mean $\rho_{SM}$ = 0.29) and terrestrial water storage tend to be strongly autocorrelated (mean $\rho_{TWS}$= 0.48). There are distinct regional differences in the annual memory of these storages (Fig. 2e-h), with stronger memory in the overall store (terrestrial water storage) than in the relatively shallower rootzone (here 250cm but note that rootzone estimates vary strongly between studies (e.g., Wang-Erlandsson et al., 2016; Stocker et al., 2023)). Rootzone soil moisture is consistently autocorrelated and associated with larger volumes of water that carry over to subsequent years (10.1 mm) compared to precipitation and potential evaporation timeseries. The strength and associated storage volumes further increase if we consider total terrestrial water storage, highlighting the role of larger long-term fluctuations than typically observed in the unsaturated zone. Terrestrial water storage has mean memory volumes of 12.7 mm.



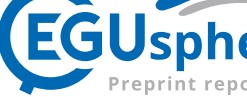





**Figure 2. Spatial and frequency distributions of annual memory in terrestrial water fluxes and stores (left column) and associated memory volumes (right column). Memory is, on average, absent in precipitation timeseries (a, b) and weak in potential evaporation (c, d). Rootzone storage (e, f) and especially terrestrial water storage (TWS) (g, h) exhibit strong memory. Outgoing fluxes still hold annual memory, both in evaporation (i, j) and in streamflow (k, l). Density plots of the autocorrelation and memory volumes also indicate that most memory, both in terms of volume and in terms of autocorrelation strength, tends to occur in terrestrial water storage (a-l).**

Outgoing fluxes, streamflow and evaporation, also exhibit substantial annual memory, although less strongly than storage. GLEAM evaporation data indicate that positive autocorrelation is also a common feature of evaporation globally (Fig. 2i) as 79% of grid cells have positive autocorrelation of which 67% is significant). The average autocorrelation value ($\rho_E = 0.16$) and associated memory values (mean $V_E$ = 5.5 mm) are typically higher than in precipitation (and other forcing) timeseries, but lower than in storage. Regional differences in GLEAM potential evaporation memory are substantially correlated with regional differences in long memory of GLEAM evaporation (Spearman correlation coefficient = 0.62). However, evaporation memory appears to be further mediated by storage, as this is also positively correlated with evaporation memory and because catchments with no potential evaporation memory still exhibit substantial evaporation memory. The estimated linear relationship between evaporation memory as a function of potential evaporation memory is: $\rho_E = 0.67\rho_{E_P} + 0.090$, which shows that also for $\rho_{E_P}$ equal zero, $\rho_E$ tends to hold memory. Regional variations in rootzone storage memory are correlated with regional differences in memory of GLEAM evaporation (Spearman correlation coefficient = 0.35; estimated linear relationship: $\rho_E = 0.25\, \rho_{SM} + 0.092$). Thus, annual memory in actual evaporation seems to be controlled both by memory in potential evaporation and by storage memory, reflecting the interplay between energy and water supply (e.g. Milly, 1994).

Streamflow records indicate that positive autocorrelation is a common feature of streamflow globally (Fig. 2k). The vast majority (79%) of catchments show signs of positive annual memory (i.e. positive autocorrelation). The average autocorrelation value ($\rho_Q = 0.16$) indicates that annual flow rates are often significantly related to their preceding year's values. The volumes of water associated with this are generally also smaller than in the storage memory volume (mean $V_Q$ = 2.7 mm) (Fig. 2l). These patterns vary regionally with regions of stronger autocorrelation (e.g. western Europe, Prairie Pothole region of the United States, Eastern Australia, most of Brazil and Russia, coastal western India), regions of weaker or negative autocorrelation (e.g., Western Australia, inland India) (Fig. 2k), and large parts of Earth's surface with few gauging stations where it remains unquantified how common annual memory is.

Annual memory in outgoing fluxes also occurs in extreme flows. Worldwide, memory is especially strong and common for annual minimum daily flows (Fig. 3a). These annual low flows have on average stronger autocorrelation (mean = 0.23) than annual mean flows (Fig 2k; Fig. 3a). This relatively strong memory likely reflects that low flows tend to be more directly sourced from groundwater-sustained baseflow and are relatively less influenced by shorter-term precipitation variability (Van Loon, 2015). Therefore, they more directly represent the strongly autocorrelated signal of terrestrial water storage. Autocorrelation is weakly present for annual maximum flows (Fig. 3b; mean = 0.07). Annual maximum flows almost always





arise through the co-occurrence of high precipitation (rainfall + snowmelt) and baseflow or soil moisture (Wasko et al., 2019; Berghuijs et al., 2016; 2019; Berghuijs & Slater, 2023). While these antecedent conditions tend to hold memory, this is less the case for precipitation, resulting in relatively weak autocorrelation in annual maximum flows globally.

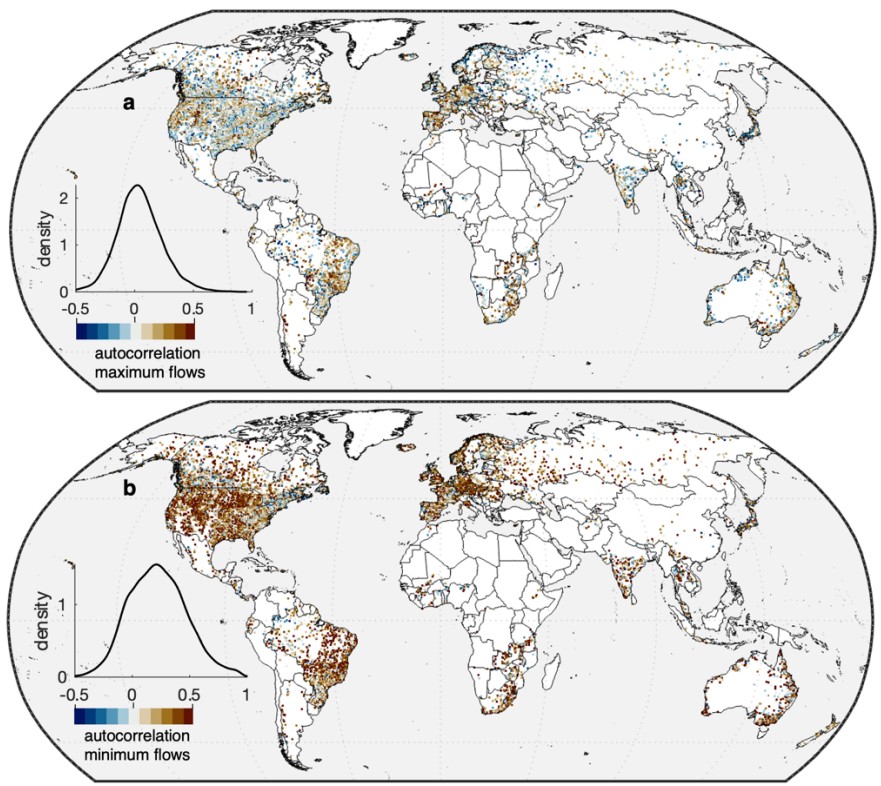

**Figure 3. Spatial and frequency distributions of memory in annual extreme flows across the study catchments. Annual memory of annual maximum (a) and minimum flows (b) is typically present but varies regionally. One-year lag autocorrelation of annual minimum flows is typically strong (mean = 0.22) compared to annual maximum flows (mean = 0.07) across the 15209 study catchments.**

**3.2 Physical controls on memory**

The spatial patterns indicate that places with stronger memory in annual mean flows also tend to have stronger memory in annual low and high-flow conditions (Fig. 2k; Fig. 3; Fig. 4). This relationship between the memory of different flow extremes is further substantiated by the Spearman rank correlation coefficients between autocorrelation of annual flows which is especially strong with that of annual minimum flows (0.54) but is also present for annual maximum flows (0.36) (Fig. 4b; note that rank correlations of the plotted binned points are visibly much stronger but vary depending on the details of the binning.

Here, each bin contains 2% of the data). The consistent variation in memory between annual mean, high, and low flows across different catchments suggests that these memory effects are not happening independently from one another but are (at least partly) underlain by similar driving mechanisms and potentially similar catchment attributes. In storage, effects of (at annual



timescales largely random) temporal variations in precipitation (and outgoing fluxes) are integrated over time (Eq. 1). The
typically randomly varying precipitation inputs integrate into more predictable storage patterns that often persist over longer
timescales (Klemeš, 1974). The strong memory of storage subsequently leads to memory of outgoing fluxes of evaporation
and streamflow.

### 3.2.1 Empirical links to catchment attributes

Correlation analysis of the memory strength with catchment attributes shows that larger catchments tend to have a substantially
stronger long memory (Fig. 4c). This scaling effect is not the only factor determining the strength of autocorrelation (also
indicated by relatively weak Spearman correlation coefficients for mean (0.09), low (0.06), and high flows (0.13)) but there is
a clear tendency of overall stronger memory for larger catchments. The 369 largest catchments in the dataset exceed sizes of
100,000 km$^2$ and cover almost 30% of Earth's land surface. In these large catchments (Fig. 5), memory, on average,
approximately doubles that of what is typical for the smallest catchments (Fig. 4c). This finding is consistent for annual
minimum, maximum, and mean flows. This growth in memory across larger catchments is empirically consistent with earlier
studies which attributed this to spatial aggregation effects (Mudelsee, 2007).

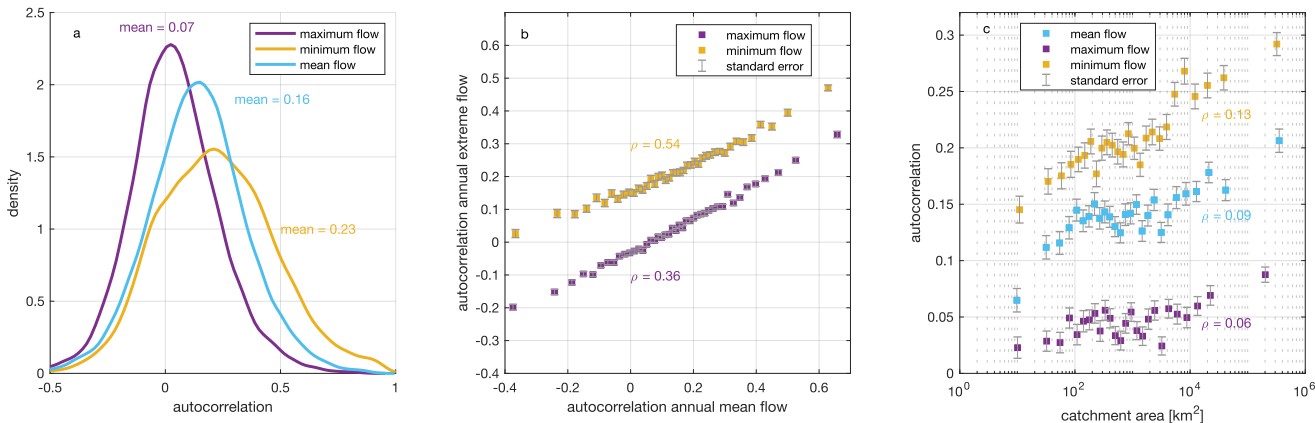

**Figure 4. Variations in annual memory are largely consistent across the flow regime and scale with catchment size. One-year lag autocorrelation differs between mean and extreme flows (a). Memory of annual extreme flows is generally larger when the same catchment also has stronger autocorrelation of annual mean flows (b) as shown by the binned plots (each bin contains 2% of the**
**data). Overall memory grows with catchment size: the largest catchments tend to have roughly twice as much memory as the smallest catchments (c). The error bars display the standard error of the mean for each bin. Spearman rank correlations ($\rho$) are shown for the unbinned values. The rank correlations of the plotted binned points are visibly much stronger but vary depending on the details of the binning.**

Spearman correlations with several other catchment properties (Do et al., 2018), such as dam numbers per unit area (0.01),
population density (-0.003), drainage density (-0.04), slope (-0.05) all exhibit weaker correlations with long memory of annual
mean flows. These results indicate that links between, for example, drainage structure and groundwater flow (e.g. Luijendijk,
2022), travel times of water and catchment slopes (e.g. Jasechko et al., 2016; Cardenas 2007) affect long memory, but
rigorously testing this requires more directed analyses, and are likely better explored at regional and local scales. Overall, from





these generally weak correlations it is hard to distinguish what catchment factors drive long memory and the results suggest
that long memory can arise across a very wide variety of catchment conditions.

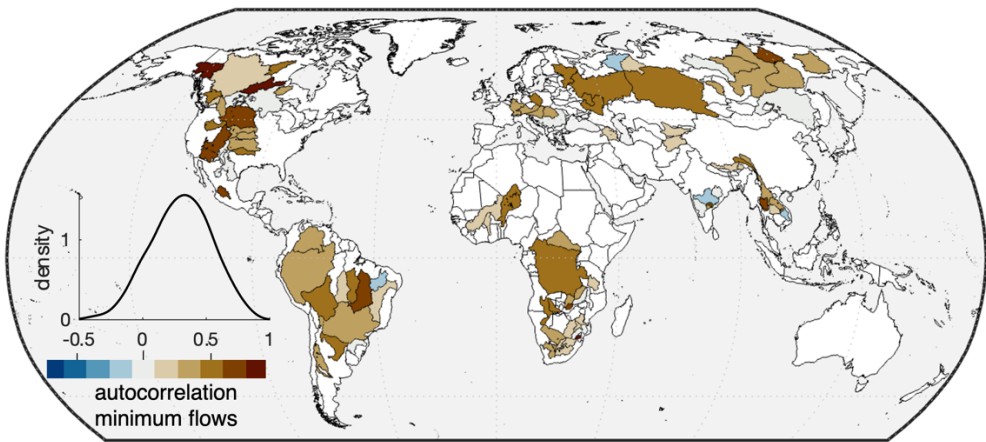

**Figure 5. Spatial and frequency distribution of memory in annual low flow rates across 369 catchments whose area exceeds 100,000 km². 88% of these larger catchments have positive autocorrelation, and this autocorrelation tends to be relatively strong on average**
**(mean = 0.30).**

### 3.2.2 Catchment functioning driving memory: model experiments

To further understand the drivers of memory in catchments, we identify that the data have shown several key memory signatures that represent typical catchment behaviour:

- Memory in annual mean streamflow arises, despite no annual memory in precipitation time series.
- Memory is stronger for annual minimum flows than for annual mean flows.
- Memory is weaker for annual maximum flow conditions than for annual mean flow
- Memory in storage is substantially stronger than in streamflow.

We seek to explain how catchments function to be consistent with these memory signatures and search for explanations
constructed with the smallest set of elements that can reproduce these signatures (i.e., we follow Occam's razor). In this search, we use synthetic model experiments that do not aim to mimic the behaviour of any particular catchment realistically, but instead, provide elementary representations of catchment functioning that capture the typical behaviour of catchments. By starting with the simplest model, we combine a top-down modelling approach (Klemeš, 1983; Sivapalan et al., 2003) with a search for model structures that are consistent with emergent behaviour across many catchments, consistent with the concept
of "functional relationships" proposed by Gnann et al. (2023). We explore various levels of model complexity (Fig. 6) and test when the model behaviour becomes broadly consistent with the observed memory signatures. If a simple model can broadly explain the signatures, we will deliberately ignore all the catchment heterogeneity and complexity that is ubiquitous in real-world catchments but that appears not to be central in the key memory behaviours.



The models we develop are forced by synthetic rainfall forcing without significant autocorrelation at annual timescales. Synthetic 500-year randomized daily rainfall timeseries vary in size at a daily scale according to a gamma distribution (shape parameter = 0.2; scale parameter = 10). In addition, annual precipitation rates vary randomly (normally distributed) with a standard deviation of 20%. We exclude, for the interests of simplicity, evapotranspiration from the models (Fig. 6). We

understand that real-world catchments evaporate part (and often even most) of their incoming precipitation (Budyko, 1974; Dorigo et al., 2021), but streamflow memory arises across a broad range of climate conditions (Fig, 2a, 3). This variety of climates, and thus long-term water balances, suggests no substantial evaporation is needed to generate long memory. In addition, evaporative fluxes can be modelled using a variety of representations (e.g., Zhao et al.., 2013; Knoben et al., 2018). Therefore, we do not explore its role in these experiments but encourage this as a potentially interesting avenue for further studies on long memory.

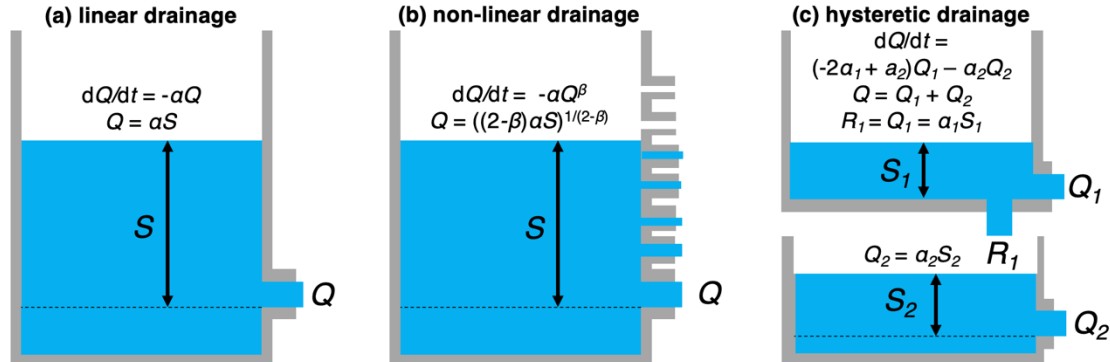

**Figure 6. Overview of catchment structures used in model experiments. The simplest model (linear drainage) (a) is expanded to include drainage nonlinearity (b), and hysteresis (c).**

**Linear drainage**

Arguably one of the simplest catchment representations is *linear drainage* (Fig. 6), where:

$$\frac{\mathrm{d}Q}{\mathrm{d}t} = -\alpha Q \tag{5}$$

$$Q = \alpha S \tag{6}$$

A linear reservoir drains water at rate $Q$ [LT$^{-1}$], directly proportional to its (effective) water storage $S$ [L] but at a rate dependent on the (fixed) drainage constant $\alpha$ [T$^{-1}$].

Logically, linear reservoirs have been reported to yield some memory effects, but these effects tend to occur at shorter than annual timescales. Many real-world catchments have linear drainage timescales of weeks (e.g., Beck et al., 2013; Botter et al., 2013; Brutsaert et al., 2008), and variations in $\alpha$ can help, for example, to discriminate erratic regimes (with enhanced intra-seasonal streamflow variability) from persistent regimes (with more regular flow patterns) (Botter et al., 2013). However, findings on the persistence of flow regimes by Botter et al. (2013) represent memory effects of the system at sub-annual

timescales. Our model experiments indicate that linear drainage does not capture annual streamflow memory, unless drainage



timescales become much longer than the typically reported timescales of weeks (Fig. 7). In addition, long memory does not vary substantially across flow conditions, and is not stronger in storage than streamflow because streamflow rates directly mirror storage conditions (Fig. 7; Eq. 8). Thus, linear drainage (unsurprisingly) fails to reproduce most of the identified key memory signatures. We also note that model experiments suggest that absolute rates of autocorrelation can be sensitive to the

forcing pattern as results vary slightly between simulations. This suggests earlier reported local autocorrelation values (e.g., Fig. 2 and 3) are not an intrinsically stable metric of catchment behaviour at the scale of an individual catchment but will vary, even when forcing statistics appear rather stable.

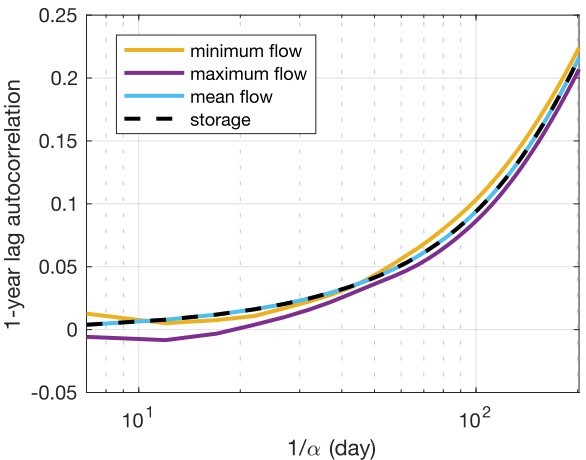

**Figure 7. Example of the scaling of drainage timescales (x-axis) and one-year lag autocorrelation (y-axis) for annual mean flows,**
**annual maximum flow, annual minimum flow, and annual mean storage for linear drainage. The slow growth of autocorrelation with drainage timescale indicates that only for much longer than typically observed drainage timescales (e.g. >100 days) substantial (>0.1) autocorrelation is generated. In addition, long memory does not vary substantially across flow conditions and is not stronger in storage than streamflow.**

**Non-linear drainage**

One step more complex catchment representation can also have *non-linear drainage* (e.g., Brutsaert and Nieber, 1977; Harman et al., 2009; Kirchner, 2009), where:

$$\frac{\mathrm{d}Q}{\mathrm{d}t} = -\alpha Q^{\beta} \tag{7}$$

$$Q = \sqrt[2-\beta]{(2-\beta)\alpha S} \tag{8}$$

Here $\beta$ [-] and $\alpha$ [$L^{1-\beta}T^{\beta-2}$] are recession constants. For $\beta$=1 drainage behaviour is linear and Eqs. 7 and 8 would become
equivalents to Eqs. 5 and 6. However, for $\beta$>1, which is commonly observed, drainage timescales become longer at low flows compared to high flows. For $\beta$<1 the opposite occurs, but this behaviour is less commonly observed in real-world catchments (e.g., Bogaart et al., 2016; Berghuijs et al., 2016). Eq. 7 can be rewritten as:

$$\frac{\mathrm{d}\hat{Q}}{\mathrm{d}t} = -a_0\hat{Q}^{\beta} \tag{9}$$





where $\hat{Q} = Q/\bar{Q}$, and $a_0$ [T$^{-1}$] is the drainage constant at the mean streamflow rate $\bar{Q}$. This allows expressing nonlinear

drainage in terms of a recession timescale $1/a_0$ [T] and drainage nonlinearity $\beta$ [-] (McMillan et al., 2014).

Our model experiments (Fig. 8) indicate that longer (normalized) recession timescales ($1/a_0$) lead to memory in annual mean streamflow (Fig. 8a), largely independent of the degree of drainage nonlinearity ($\beta$). In addition, these simulations show stronger memory in low flows than in mean and high flows (Fig. 8a-c), whereby the low flows exhibit more memory when

drainage becomes more nonlinear whereas high flows show decreasing memory under more nonlinear drainage. Memory in storage and streamflow is still near equivalent (Fig. 8d). The latter is shown by differences between annual mean storage and annual mean flow all being very close to zero (Fig. 8d). Combinations of $\alpha$ and $\beta$ that are derived from streamflow timeseries across many catchments (e.g., Bogaart et al., 2016; Berghuijs et al., 2016) tend to be mostly outside the range expected to yield substantial annual memory. These inferences suggest that drainage non-linearity by itself is not yet a mechanism that solely

can be responsible for all the observed long memory signatures.

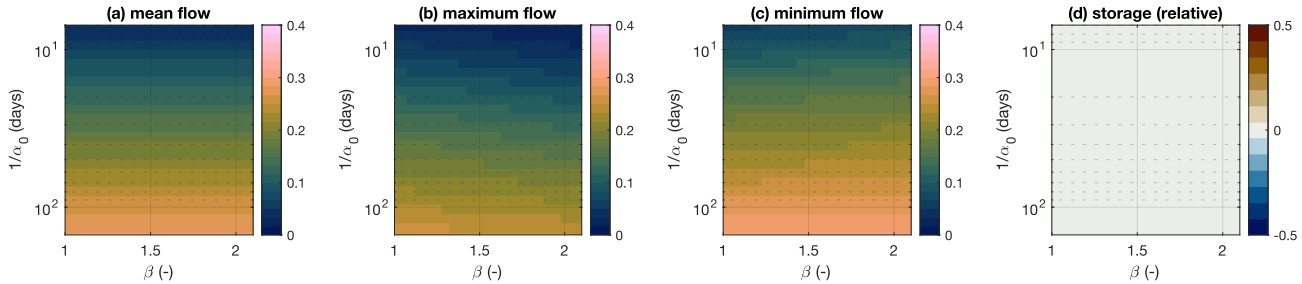

**Figure 8. Memory generated in nonlinear drainage conditions as a function of the normalized drainage timescale ($1/\alpha_0$) and the drainage nonlinearity ($\beta$). Colors indicate 1-year lag autocorrelation for annual mean flows (a), annual maximum flow (b), annual minimum flow (c), and the memory of annual mean storage minus the memory of annual mean flows (d) which shows no substantial**

**differences. The latter emphasizes that there is no clear memory difference between storage and streamflow. Note that y-axes displaying the drainage timescale are logarithmic.**

### Hysteretic drainage

Non-linearity can cause some annual memory (Fig. 8) but non-hysteretic catchment representations (i.e. with a unique storage-discharge relationship) used this far cause storage and outflows to be similarly autocorrelated (Fig. 8d). A model of two (in

series) connected reservoirs, that both linearly drain water but at two different timescales results in *hysteretic drainage* (e.g., Fovet et al., 2015; Gharari et al., 2018), where:

$$\frac{dQ}{dt} = (-2\alpha_1 + a_2)Q_1 - \alpha_2 Q_2 \tag{10}$$

$$Q = Q_1 + Q_2 = \alpha_1 S_1 + \alpha_2 S_2 \tag{11}$$

Drainage timescales of the upper reservoir (parameterized by $\alpha_1$ [T$^{-1}$]) broadly reflect (typically faster) runoff generation in

and over the unsaturated zone, whereas drainage timescales from the lower reservoir (parameterized by $\alpha_2$ [T$^{-1}$]) broadly reflect (typically slower) groundwater drainage. Recharge $R_1$ [LT$^{-1}$] from the upper box (where all precipitation enters) towards



the lower box is also set at a rate $Q_1$ [LT$^{-1}$] which means that half of the drainage from the upper reservoir passes through the lower reservoir before becoming streamflow. This simplified relationship is broadly consistent with estimates that just over half of global river flow originates from groundwater (Xie et al., 2024).


In this hysteretic (and thus non-linear) drainage setup, a catchment can have a fast streamflow response in the upper reservoir while also exhibiting an underlying lower frequency streamflow variation driven by the lower reservoir. Half of the drainage from the upper reservoir passes through the lower reservoir before becoming streamflow, therefore, long-term averages of $Q_1$ and $Q_2$ must be similar. This also implies that the mean storage volumes of the two reservoirs are related to one another according to the ratio of the two drainage timescales: $\overline{S_2} \approx \frac{\alpha_1}{a_2}\overline{S_1}$. Thus, when the lower reservoir has a substantially longer drainage timescale than the upper reservoir, most water will be stored in the lower reservoir and this storage will vary more slowly than the upper reservoir. Consequently, in this setup, memory can become substantially stronger in overall storage ($S_1$+$S_2$) than in streamflow ($Q_1$+$Q_2$).

Our model experiments indicate that, especially for sufficiently long drainage timescales of the lower reservoir, streamflow can exhibit annual memory in streamflow (Fig. 9a). This memory in annual mean flows is largely independent of the faster-draining upper reservoir. When the drainage in the upper reservoir is much faster than in the lower reservoir, several other simulated memory signatures also become consistent with the observations. Namely, memory becomes stronger for annual low flows than annual mean flows (Fig. 9a-c) because low flows are more determined by the slower lower reservoir. In addition, memory is much weaker for annual maximum flow conditions than annual mean flow, because high flows are more determined by the faster upper reservoir (which holds less memory). Furthermore, annual memory in storage is substantially stronger than in streamflow because most storage is present in the lower reservoir (Fig. 9d), and this storage varies across longer timescales than overall streamflow (of which only 50% originates from this slower reservoir).

Our simulations suggest that hysteresis is central to the observed memory signatures. We acknowledge that these simulations are overly simplified catchment representations and not a detailed and accurate model for any specific catchment. However, they demonstrate how some basic mechanisms (which are also encodable in more complex and detailed models) can lead to behaviour that is consistent with the observed memory signatures. In addition, the model outcomes also reflect those of detailed model implementations in real-world catchments where the addition of a groundwater component that allowed for very low-frequency fluctuations in groundwater flows helped to simultaneously improve streamflow and water storage predictions substantially (e.g., Hulsman et al., 2021).



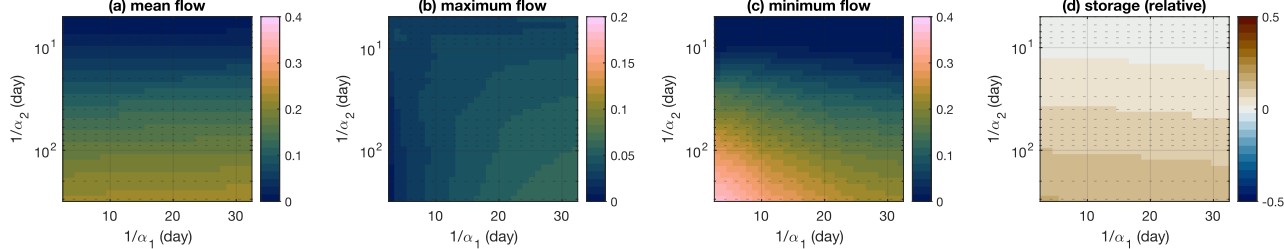

**Figure 9. Memory generated in hysteretic drainage conditions. The axes indicate variations in the drainage timescales of both reservoirs whereas colors indicate 1-year lag autocorrelation for annual mean flows (a), annual maximum flow (b), annual minimum flow (c), and the memory of annual mean storage minus that of annual mean flows (d). Note that y-axes are logarithmic.**

## 4. Conclusions

Long memory has been discovered decades ago in annual flow records of the River Nile and later more widespread evidence of long memory in annual flows has been reported for many rivers worldwide. The presence of such memory could prolong drought and flood risks and affect water resources over long periods, but the global universality, strength, and origin of long memory in the water balance components and hydrological extremes remained largely unquantified. Here, we quantified annual memory in the terrestrial water cycle using autocorrelation applied to annual time series of water balance components globally. The global gridded and catchment-scale datasets used here indicate that annual memory is typically weak in incoming terrestrial fluxes and forcing but becomes strong in terrestrial water stores, and cascades into outgoing fluxes. Annual memory is not limited to annual streamflow rates but often extends toward annual extreme flows and is especially strong for low flows and in large catchments. Our model experiments indicate that this memory arises with increasing nonlinearity of catchment response, but storage-discharge relationships also need to account for hysteresis effects to produce all observed memory signatures. Incorporating these dynamics may be important to produce multi-year low-frequency variations in the terrestrial water cycle as recently demonstrated by Hulsman et al. (2021) and could also have implications for our understanding of other processes that may be affected by memory of water stores and fluxes such as vegetation dynamics (Koirala et al., 2017) and atmospheric CO2 growth rates (Humphrey et al., 2018).

**Data Availability Statement** Data is publicly available via the cited sources.

**Author Contributions** W.R.B. initiated the study, performed all analyses, and led the writing. All authors contributed to refining analyses and writing.

**Conflict of Interest** At least one of the (co-)authors is a member of the editorial board of Hydrology and Earth System Sciences.



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
