# Peer review of "Annual memory in the terrestrial water cycle"

_EGUsphere, 2024_

## Referee Comment (RC1)

Title:   Annual memory in the terrestrial water cycle

Authors: Berghuijs et al.

**Review**

This manuscript describes an initial evaluation of autocorrelation in catchment water balance components on a global basis.

The topic is highly suitable for the journal.

The manuscript is interesting, very well written and easy to follow.

I had very few substantive comments and the paper can be published more or less as is.

**Details**

1.  Line 53. I was pleased to see the caveat about inter-basin flows. I suspect this could be extended to land-ocean transfers as well. (If you are curious, google "wonky holes" and have a read. I have personally drunk fresh water over the side of a boat 50 km from land on the Great Barrier Reef. The local fisherman have known this for a long long time.)

2.  Line 107. Typo? Should it be Sun et al 2018 (and not 2017) or is there another reference?

3.  Line 114. Perhaps ….. Thus $\rho_y$ **roughly** expresses. Or "**approximately**" instead of roughly if you like but you need a qualifier here.

4.  Figure 2e. Can you speculate on a likely physical explanation for the negative autocorrelation values, that occur in different climates, e.g. semi-arid South Western Australia, Botswana, bottom of South America, and in the cold parts of northern Russia and in other widely varying climates?

5.  Figure 2eik. Focus on South Western Australia. You have autocorrelations as follows; -ve for rootzone (Fig 2e), -ve for evaporation (Fig 2i) and 0 for annual flow (Fig. 2k). Makes sense since in that region there is minimal streamflow and I would interpret this as a good plant growing year (enhanced evaporation) depletes rootzone moisture. But I could not imagine how that would work in northern Russia with the same spatial patterns as above. No change requested but I was intrigued. (I had personally imagined a study like this for years and am glad that it has now been done.)

6.  Line 339. Typo? .. used **thus** far cause

**Michael L. Roderick, 9/10/2024**

---

## Referee Comment (RC2)

[referee-annotated manuscript omitted]

---

## Author Comment (AC1)

Dear Dr. Roderick,

Thank you for this constructive review.  Please find our responses below (**in bold**).

Best regards,

Wouter Berghuijs (on behalf of all authors)
* * *
This manuscript describes an initial evaluation of autocorrelation in catchment water balance components on a global basis.

The topic is highly suitable for the journal.

The manuscript is interesting, very well written and easy to follow.

I had very few substantive comments and the paper can be published more or less as is.

**We appreciate this constructive review.**

Line 53. I was pleased to see the caveat about inter-basin flows. I suspect this could be extended to land-ocean transfers as well. (If you are curious, google "wonky holes" and have a read. I have personally drunk fresh water over the side of a boat 50 km from land on the Great Barrier Reef. The local fisherman have known this for a long long time.)

**In the revised manuscript, we will also mention land-ocean transfers. I was unaware of *wonky holes*, but they provide an excellent illustration of why land-ocean transfers should be mentioned.**

Line 107. Typo? Should it be Sun et al 2018 (and not 2017) or is there another reference?

**It indeed should be 2018.  In the revised manuscript, we will correct this.**

Line 114. Perhaps ….. Thus $\rho y$ **roughly** expresses. Or "**approximately**" instead of roughly if you like but you need a qualifier here.

**In the revised manuscript, we will add the qualifier "approximately".**

Figure 2e. Can you speculate on a likely physical explanation for the negative autocorrelation values, that occur in different climates, e.g. semi-arid South Western Australia, Botswana, bottom of South America, and in the cold parts of northern Russia and in other widely varying climates?

**In the revised manuscript, we will speculate on the potential physical (and statistical) causes of this negative autocorrelation.**

Figure 2eik. Focus on South Western Australia. You have autocorrelations as follows; -ve for rootzone (Fig 2e), -ve for evaporation (Fig 2i) and 0 for annual flow (Fig. 2k). Makes sense since in that region there is minimal streamflow and I would interpret this as a good plant growing year (enhanced evaporation) depletes rootzone moisture. But I could not imagine how that would work in northern Russia with the same spatial patterns as above. No change requested but I was intrigued. (I had personally imagined a study like this for years and am glad that it has now been done.)

**In the revised manuscript, we will briefly discuss the potential causes.**

Line 339. Typo? .. used **thus** far cause

**We will correct this.**

---

## Author Comment (AC2)

Dear Reviewer,

Thank you for this constructive review. Please find our first responses below (**in bold**).

Best regards,

Wouter Berghuijs (on behalf of all authors)
* * *
This is a very interesting paper on the long-term memory in precipitation, evaporation, streamflow and storage. The analyses are clearly described. The writing is clear and the figures are all very informative. I have no major comments and highly recommend publication of the manuscript in HESS.

**Thank you.**

My main comment is related to the structure of the paper. The model part is mentioned at the end of the introduction but is not part of the methods and almost came as a surprise to me. I would move some of the parts of the model section into the methods (e.g., the analyses and the description of the three model structures) and then divide the results into those related to the data analyses and those related to the model results. That way, a) the model section has a better division of methods and results/discussion, and b) the model part is part of the methods and doesn't appear to almost come as an afterthought. The model results are very interesting but I think that they could use a bit more discussion, e.g., it would be useful to highlight which lumped bucket type models have the tested structures, to highlight that this type of analysis may help to determine what type of model structure one needs to use, and that it means that any model structure testing with short datasets needs to be done with care!

**In the revised manuscript, we will extend the modeling part in the methods section. In addition, we will extend the discussion on our modeling and its results.**

My other comment relates to the datasets used. I fully agree with the choice of the datasets for ET and storage but it would be useful if there was at least some critical reflection of the datasets. Afterall, there is some "modelling" already involved in getting the "data". Thus, as with any data, there are some uncertainties in the data. There is currently no discussion on how this may influence the outcomes.

**In the revised manuscript, we will provide critical reflections on the datasets and our use of these data.**

Minor suggestions –really just suggestions:

- L98: Explain why you left out these arctic areas, rather than only stating that you left them out.

**In the revised manuscript, we will clarify this.**

- L126, L188: Considering all uncertainties, I would not include the decimal.

**In the revised manuscript, we will not include this.**

- L173: Is this 67% of the 79% or 67% of all the pixels? This could be worded more clearly.

**In the revised manuscript, we will clarify that this is 67% of all pixels.**

- L176, L181: Replace 'Spearman rank coefficient' by a symbol, but ideally not rho (see comment below).

**In the revised manuscript, we will use $r_s$ for the Spearman rank correlation**

- L195, L199: Add symbol after 'mean' for greater clarity.

**In the revised manuscript, we will add this.**

- L209: I don't think that the header is very fitting to the contents of the section. In fact, I think that you can just leave the header out and include the text as a continuation of the previous section.

**This header covers the entire 3.2 section, including empirical analyses (3.2.1) and model experiments (3.2.2), both addressing the physical causes of memory. Therefore, we do not think that removing this header is beneficial**

- L226: It would be helpful for the readers if you gave your thoughts on why the larger catchments have a stronger long-term memory. Is it the presence or importance of larger (alluvial) aquifers? The fact that there are likely more (large) lakes for larger catchments? Or that larger catchments are overall flatter?

**In the revised manuscript, we will discuss the potential causes of stronger memory in larger catchments.**

- L236: I found it a bit confusing that rho is use for both the memory ($r_y$) and the Spearman rank correlation (r). Consider using $r_s$ for the Spearman rank correlation instead.

**In the revised manuscript, we will use $r_s$ for the Spearman rank correlation**

- L239-240: Are these very low correlations statistically significant?

**In the revised manuscript, we will report these (significant) *p*-values**

---

## Author Response (AR1)

Dear Editor,

We appreciate the reviewers' feedback and have provided detailed responses below.

In addition to the reviewer comments, and following a suggestion from Prof. James Kirchner, we modified the memory volume to be multiplied by the standard deviation of the annual anomaly instead of the mean absolute anomaly. This change did not affect the overall conclusions drawn from the results.

Best regards,

Wouter Berghuijs (on behalf of all authors)
* * ** * *
**RESPONSE REVIEW 1**

This manuscript describes an initial evaluation of autocorrelation in catchment water balance components on a global basis.

The topic is highly suitable for the journal.

The manuscript is interesting, very well written and easy to follow.

I had very few substantive comments and the paper can be published more or less as is.

**Response: We appreciate this constructive review.**

**Changes to manuscript: None**

Line 53. I was pleased to see the caveat about inter-basin flows. I suspect this could be extended to land-ocean transfers as well. (If you are curious, google "wonky holes" and have a read. I have personally drunk fresh water over the side of a boat 50 km from land on the Great Barrier Reef. The local fisherman have known this for a long long time.)

**Response: In the revised manuscript, we now also mention land-ocean transfers.**

**Changes to manuscript: We now state: *"(ignoring potential inter-basin groundwater flows or land-ocean transfers)"***

Line 107. Typo? Should it be Sun et al 2018 (and not 2017) or is there another reference?

**Response: indeed should be 2018.**

**Changes to manuscript: We now state: *Sun et al. (2018)***

Line 114. Perhaps ….. Thus $\rho y$ **roughly** expresses. Or "**approximately**" instead of roughly if you like but you need a qualifier here.

**Response: agreed**

**Changes to manuscript: we added the qualifier "*approximately*".**

Figure 2e. Can you speculate on a likely physical explanation for the negative autocorrelation values, that occur in different climates, e.g. semi-arid South Western Australia, Botswana, bottom of South America, and in the cold parts of northern Russia and in other widely varying climates?

**Response: We now speculate on this negative autocorrelation's potential physical (and statistical) causes.**

**Changes to manuscript: Areas with negative autocorrelation in rootzone storage frequently overlap with regions of negatively autocorrelated precipitation and grasslands. We hypothesize that the limited rootzone storage capacity in these regions is insufficient to buffer against variability in precipitation patterns. As a result, the temporal variability of precipitation is mirrored in the rootzone storage, allowing the precipitation's negative autocorrelation to propagate through the system.**

Figure 2eik. Focus on South Western Australia. You have autocorrelations as follows; -ve for rootzone (Fig 2e), -ve for evaporation (Fig 2i) and 0 for annual flow (Fig. 2k). Makes sense since in that region there is minimal streamflow and I would interpret this as a good plant growing year (enhanced evaporation) depletes rootzone moisture. But I could not imagine how that would work in northern Russia with the same spatial patterns as above. No change requested but I was intrigued. (I had personally imagined a study like this for years and am glad that it has now been done.)

**Response: We agree this is an interesting pattern but causes are likely complex and beyond the scope of what we feel we can reasonably speculate on at this stage.**

**Changes to manuscript: None**

Line 339. Typo? .. used **thus** far cause

**Response: agreed**

**Changes to manuscript: we now state "*thus*".**

\*\*\*\*\*\*
**RESPONSE REVIEW 2**

This is a very interesting paper on the long-term memory in precipitation, evaporation, streamflow and storage. The analyses are clearly described. The writing is clear and the figures are all very informative. I have no major comments and highly recommend publication of the manuscript in HESS.

**Response: Thank you.**

**Changes to manuscript: None**

My main comment is related to the structure of the paper. The model part is mentioned at the end of the introduction but is not part of the methods and almost came as a surprise to me. I would move some of the parts of the model section into the methods (e.g., the analyses and the description of the three model structures) and then divide the results into those related to the data analyses and those related to the model results. That way, a) the model section has a better division of methods and results/discussion, and b) the model part is part of the methods and doesn't appear to almost come as an afterthought.

**Response: We agree that the paper has a somewhat unconventional division between sections, but this is done on purpose to improve the logical flow of the work. We have made a few amendments that balance using a more traditional set-up, but that do not destroy the logic we already presented. Already presenting model structures for a top-down modeling experiment would become illogical in our opinion.**

**Changes to manuscript: In the methods section of the manuscript we now state:**

**"2.3 Model experiments**

**We test how catchments can function to be in alignment with empirically derived memory behaviours. Our model experiments seek the most compact representation capable of replicating emergent behaviour across the many catchments. We examine different levels of model complexity**

**and evaluate when the model's behaviour aligns broadly with the observed memory signatures. These model experiments, which rely on the empirical results, are discussed in detail in Section 3.2.2."**

The model results are very interesting but I think that they could use a bit more discussion, e.g., it would be useful to highlight which lumped bucket type models have the tested structures, to highlight that this type of analysis may help to determine what type of model structure one needs to use, and that it means that any model structure testing with short datasets needs to be done with care!

**Response: We now reflect on model structures that use bucket type models.**

**Changes to manuscript: In section 3.2.2 we state: "Bucket-type spatially lumped models are widely employed in catchment modelling (for a good overview see: Knoben et al., 2019). More complex (spatially distributed) model structures could also be explored to examine their degree of nonlinearity and hysteresis."**

My other comment relates to the datasets used. I fully agree with the choice of the datasets for ET and storage but it would be useful if there was at least some critical reflection of the datasets. Afterall, there is some "modelling" already involved in getting the "data". Thus, as with any data, there are some uncertainties in the data. There is currently no discussion on how this may influence the outcomes.

**Response: In the revised manuscript, we reflect on the datasets and our use of these data.**

**Changes to manuscript: Before the modeling experiments we now state:**

**"The empirical patterns presented (Figs. 2-5) are subject to observational and model uncertainties. Global precipitation datasets often contain significant uncertainties, which propagate to derived data products such as soil moisture storage and evaporation (Khan et al., 2018). Storage time series from GTWS-MLrec combine GRACE observational time series with machine learning models—partially trained on meteorological data—to extend terrestrial water storage estimates to periods preceding satellite observations (Yin et al., 2023). Streamflow time series, while relatively independent indicators of long-term hydrological variability, may also partially reflect processes like riverbed aggradation and degradation (Slater et al., 2019), which likely exhibit long-term memory effects. As a result, local-scale memory behavior is likely to carry considerable uncertainty. In the following sections, we concentrate on the broader patterns that emerge across multiple catchments, which are likely to be largely unaffected by local data uncertainties."**

Minor suggestions –really just suggestions:

• L98: Explain why you left out these arctic areas, rather than only stating that you left them out.

**In the revised manuscript, we state: We exclude data from Greenland, Iceland, and Antarctica from our analysis due to their unique climatic and environmental conditions, which differ significantly from other regions**

• L126, L188: Considering all uncertainties, I would not include the decimal.

**Response: no decimals found at L126, and no changes made to L188**

• L173: Is this 67% of the 79% or 67% of all the pixels? This could be worded more clearly.

**In the revised manuscript, we state "whereby 67% of all cells are significant"**

• L176, L181: Replace 'Spearman rank coeffiicient' by a symbol, but ideally not rho (see comment below).

**Agreed. We now use $r_S$ for the Spearman rank correlation.**

• L195, L199: Add symbol after 'mean' for greater clarity.

**Agreed: symbols are added**

• L209: I don't think that the header is very fitting to the contents of the section. In fact, I think that you can just leave the header out and include the text as a continuation of the previous section.

**This header covers the entire 3.2 section, including empirical analyses (3.2.1) and model experiments (3.2.2), both addressing the physical causes of memory. Therefore, removing this header is not beneficial.**

• L226: It would be helpful for the readers if you gave your thoughts on why the larger catchments have a stronger long-term memory. Is it the presence or importance of larger (alluvial) aquifers? The fact that there are likely more (large) lakes for larger catchments? Or that larger catchments are overall flatter?

**We now state: "Larger catchments often have more substantial alluvial aquifers, potentially enhancing memory effects."**

**We do not discuss slope or dam density here as these are tested for later (and appear to be weaker controls).**

• L236: I found it a bit confusing that rho is use for both the memory ($r_y$) and the Spearman rank correlation (r). Consider using $r_S$ for the Spearman rank correlation instead.

**Agreed. We now use $r_S$ for the Spearman rank correlation.**

• L239-240: Are these very low correlations statistically significant?

**We now report *p*-values**